# Studying the Construction of Floor Mosaics in the Roman Villa of Pisões (Portugal) Using Noninvasive Methods: High-Resolution 3D GPR and Photogrammetry

**Bento Caldeira [1],\***, **Rui Jorge Oliveira [2]**, **Teresa Teixidó [3]**, **José Fernando Borges [1]**, **Renato Henriques [4]**, **André Carneiro [5]** and **José Antonio Peña [3]**

1   Institute of Earth Sciences/Physics Department, University of Évora, Rua Romão Ramalho, 59, 7002-554 Évora, Portugal
2   Institute of Earth Sciences, University of Évora, Rua Romão Ramalho, 59, 7002-554 Évora, Portugal
3   Andalusian Institute of Geophysics, University of Granada, Campus Universitario de Cartuja, 18071 Granada, Spain
4   Institute of Earth Sciences, University of Minho, Campus de Gualtar, 4710-057 Braga, Portugal
5   Centre for Art History and Artistic Research/History and Archaeology Department, University of Évora, Largo dos Colegiais, 2, 7000-803 Évora, Portugal
\*   Correspondence: bafcc@uevora.pt

**Abstract:** Over the past decade, high-resolution noninvasive sensors have been widely used in explorations of the first few meters underground at archaeological sites. However, remote sensing actions aimed at the study of structural elements that require a very high resolution are rare. In this study, layer characterization of the floor mosaic substrate of the Pisões Roman archaeological site was carried out. This work was performed with two noninvasive techniques: 3D ground penetrating radar (3D GPR) operating with a 1.6 GHz central frequency antenna, which is a very high-resolution geophysical method, and photogrammetry with imagery obtained by an unmanned aerial vehicle (UAV), which is a very high-resolution optical method. The first method allows penetration up to 30–40 cm depth and 3D models can be obtained, and with the second method, very high detail surface images and digital surface models can be obtained. In this study, we analyze a combination of data from both sensors to study a portion of the floor mosaic of the Pisões Roman Villa (Beja, Portugal) to obtain evidence of the inner structure. In this context, we have detected the main structural levels of the Roman mosaic and some internal characteristics, such as etched guides, internal cracking, and detection of higher humidity areas. The methodology that we introduce in this work can be referenced for the documentation of ancient pavements and may be used prior to carrying out preservation activities. Additionally, we intend to show that a Roman mosaic, understood as an archaeological structure, does not consist of only beautiful superficial drawings defined by the tesserae, but these mosaics are much more complex elements that must be considered in their entirety for preservation.

**Keywords:** subsurface detection; GPR and photogrammetry UAV; Roman mosaic structure; noninvasive archaeological inspection

## 1. Introduction

The Roman Villa of Pisões (Figure 1) is located 10 km south of Beja (Portugal). This villa was discovered accidentally in 1967 and is considered by the academic and scientific community to be a heritage site of the greatest cultural relevance and was classified as a property of public interest in

1970. The excavations made during the 1970s revealed a pars urbana (residential house) of a singular monument of the Roman province of Lusitania composed of more than 40 divisions, distributed around a small peristyle in the tradition of the Italian atrium, which is gathered around the four-colonnaded space (Figure 1). The south facing, large porticoes with columns open to face a large water mirror with a length of 40 m, one of the largest of this type in private residences throughout the Iberian Peninsula [1]. The architectural design and type of structures [2], refinement of the decoration, especially the great quality of the floor covering materials, mosaics and marbles [3,4], bring to mind a luxurious residence with proven occupation between the 1st and 6th centuries A.D. [1]. In addition to the residence, the thermal baths are considered to be the most relevant examples of private baths [5], which are remarkably well-preserved in some compartments, with an entire heating system that allowed the circulation of hot air between double walls and under the pavements of the rooms. This complex system of brick arched roof, known as hypocaust, is exceptionally well preserved [6].

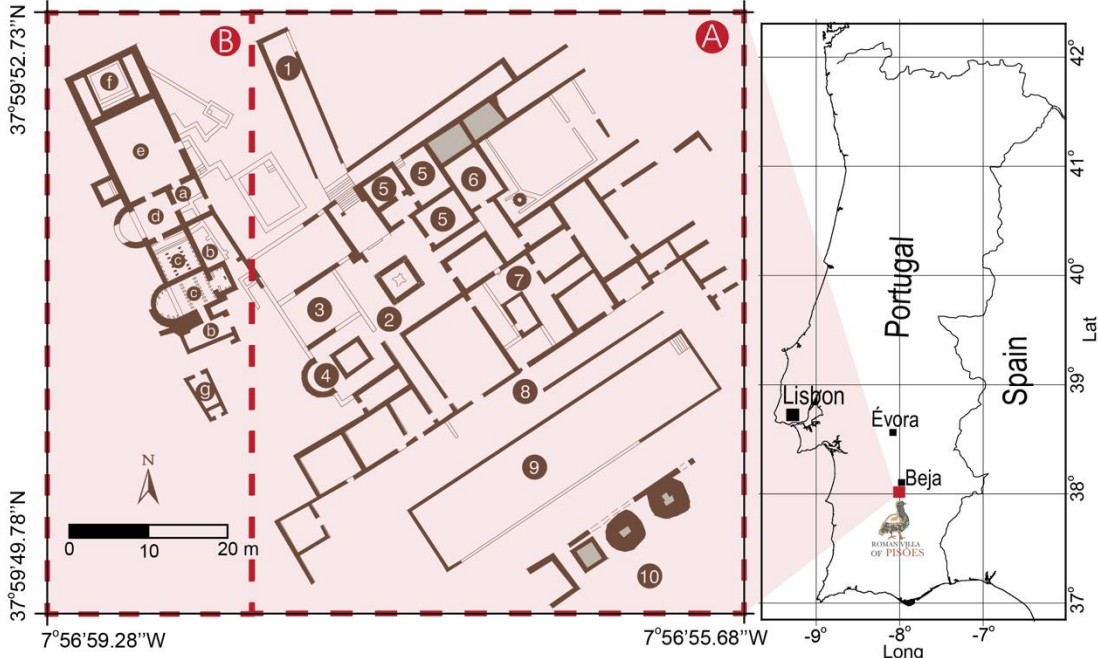

**Figure 1.** Localization of the Roman Villa of Pisões in Portugal map, and the plan view of the pars urbana with the (**A**) residential area and (**B**) thermal baths. 1. North entrance corridor, where the studied mosaics are located; 2. columned atrium; 3. reception room (tablinium); 4. dining room (triclinium); 5. bedrooms; 6. larder and kitchen; 7. former atrium with square basin for rainwater (implivium); 8. south porticoe; 9. swimming pool (natatio); and 10. mausoleums. In the baths: a. Dressing room (apodyterium); b. furnace (praefurnium); c. hot water room (caldarium); d. warm room (tepidarium); e. cooling room (frigidarium); f. swimming pool (natatio); and g. latrines.

Despite the relevance attributed to this villa, the pars rustica and pars fructuaria remain underground and undiscovered. The knowledge about the excavated areas remains poor due the scarcity of documentation resulting from the excavations made at that time, a lack of stratigraphic sequences, and the absence of other studies performed thereafter.

In 2017, the University of Évora assumed the tutelage of the villa with the commitment to promote multidisciplinary scientific research to study, preserve, value, and disseminate that heritage [7]. Thus, the investigation returned to Pisões, and the ongoing work aims to conduct a multidisciplinary research approach based on two essential pillars: a) Analysis and interpretation of structures and artifacts already known and b) research of underground structures. Due to the fact that the preservation aspect guides the dynamics of this plan, all activities are defined to develop, refine, and apply research

methodologies that minimize impacts, rigorously reproduce the geometric and functional memory of the structures, and ensure the scrupulous preservation of all data collected throughout the whole site.

Archaeological geophysics are noninvasive survey techniques that are sometimes used for the analysis of architectural elements and artifacts of the Roman villae [8,9], but in the Roman villae located in Lusitania, these methodologies have rarely been used. The study of Roman material culture found in archaeological excavations still mostly relies on typological and comparative approaches. However, in recent years, the approaches in this field have changed with the introduction of archaeometric and geophysical analyses to promote the inspection of structures prior to interventions of excavation or conservation, e.g., [10,11], and to contribute to a better understanding of the site and works of the people who lived in that villa.

Unmanned aerial photogrammetric survey is a technique widely used in the study and documentation of Roman mosaics [12–14], but to date, there are few bibliographic examples that employ geophysical methods to inspect Roman mosaics. Most of these works use the ground penetrating radar method (GPR) as the better technique to obtain detailed information on these structures. That said, all of them focus their studies on evaluating the degree of conservation and the humidity concentration through the amplitude reflection analysis in 2D-GPR profiles [15,16] or consider horizontal slices of 40 cm thickness in 3D-GPR sets to detect anomalies within a wall covered by mosaic [17]. In this context, the present study is the first where a Roman mosaic is investigated to know how it has been constructed and what its internal stratigraphy is. Effectively, the analysis of man-made materials such as stones, bricks, mortars, marble, and mosaics can provide significant clues to resolve archaeological issues, shedding light on topics related to the raw material provenance and their structural characterizations, thus deepening our knowledge of the technology used by the Romanized population that lived at Pisões.

Since the discovery of Pisões, one of the more relevant issues associated with it is the abundance of leisure elements and the quality of the covering materials, especially the collection of floor mosaics. In an interview with "Diário de Notícias" (national newspaper) on February 21, 1967, which was a few days after the discovery, the archaeologist stated, "In the soil observed so far, we have found two species of mosaics, one of *opus signinum* and the other with geometric designs in white, dark blue and red . . . ".

Later, the excavations uncovered mosaics with figures of nature and geometrics drawings in approximately 40 division floors of the urban pars. This collection of mosaics is notable, both for its eclectic nature and richness of iconology, presenting geometric and naturalistic compositions, and for the quality of its execution, from the oldest monochromaticity to the most recent polychromaticity [6,18].

In this paper, the constituting layers characterization of the floor mosaic substrate from the Roman archaeological site of Pisões was carried out. The work, completed using noninvasive geophysical and photogrammetric methods, allowed us to obtain evidence of the construction process, which is adequate as a model for the documentation of ancient pavements and can be used prior to conservation actions. Additionally, we intend to show that a Roman mosaic, understood as an archaeological structure, does not consist of only beautiful superficial drawings defined by the tesserae, but is a much more complex element that must be considered in its entirety.

## 2. Materials and Methods

### 2.1. Manufacturing Processes of the Roman Floor

The term mosaic is associated with the presence of a tesserae and represents the result of a mixing of arts and techniques from different parts of the Roman Empire, and mosaics are used as decorative elements in many environments [19]. The different aspects related to artistic styles or historical cultural issues represented on the surface of the mosaics is widely documented. However, when we look for the invisible structure that demonstrates how the substrate of the well-studied mosaics was formed, the information is scarce or absent. Concerning this question, practically all works cite the treatise of

Architecture of Vitruvius, De Architectura, written in the 1st century B.C. and translated into nearly all languages [20]; the author documents the techniques used for the preparation of mosaic pavements, highlights the impermeability issue of the mosaic in terms of functionality, and supplies the ideal method to ensure impermeability. Summarizing Vitruvius's rules, the floor mosaic foundation should be manufactured with the following stratigraphy (Figure 2):

(1)　The first layer, called statumen, is made of stones approximately 12 cm thick "each of which is not less than a handful", laid vertically on the ground, without mortar between the stones, which gives stability to the pavement and favors the flow of infiltrated waters.

(2)　The second layer, called rudus, is spread over the statumen and consists of a mortar of sand and gravel (three quarters) mixed with lime (a fourth part). This layer is rammed by means of wooden stamps until a unified level is formed, which is then left to dry. The thickness of this layer is at least 22 cm.

(3)　The third layer, called the nucleus, is a ceramic mortar that is 11–15 cm thick, made of powdered pottery and lime in proportions of three to one, respectively, and this mortar is spread over the rudus and allowed to dry. This layer is the hardest and most impermeable of the three layers. The purpose of this layer was to allow drainage of surface waters.

(4)　On top of the nucleus, a bedding layer of mortar very rich in lime is thinly applied in small sections over the nucleus. This thin layer is applied according to the forecast of the working day and fresh enough to mark the reference points of the drawing (sinopia) or geometric or figurative design. Tesserae are inserted in this layer before the mortar dries.

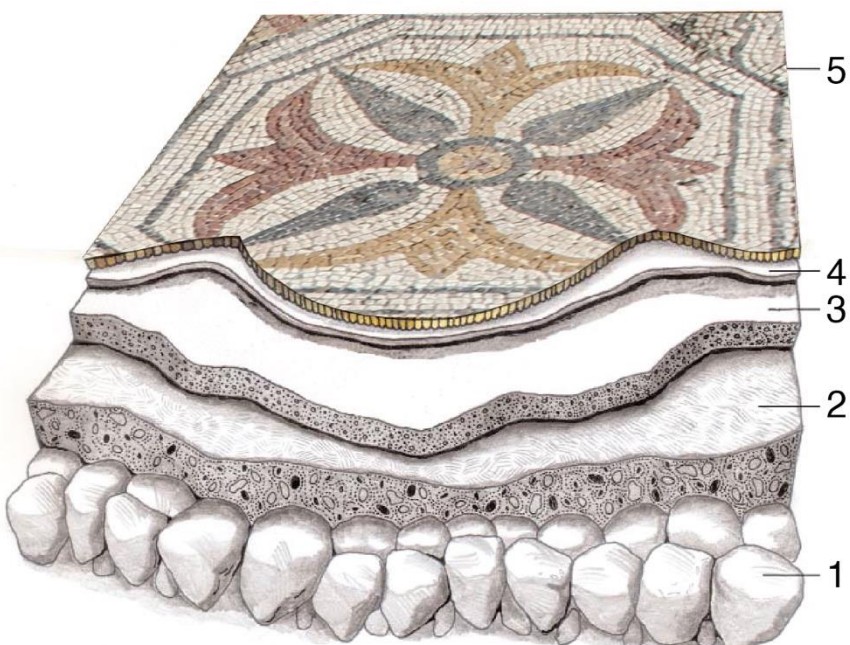

**Figure 2.** Stratigraphy of a Roman mosaic floor according to Vitruvius's description. The scheme is adapted from the Getty Conservation Institute and the Israel Antiquities Authorities and shows a fragment of the studied mosaic. **1**. Statumen—First preparatory layer of large stones; **2**. Rudus—mortar layer of gravel to level the floor; **3**. Nucleus—impermeable mortar layer of powdered pottery and lime to protect the mosaic surface from the soil water (humidity); 4. Bedding—thin mortar layer, rich in lime, applied fresh in small regular portions where the mosaic designs were marked, and the tesserae were inserted before the mortar dried; and **5**. Tessellatum—surface of the mosaic composed of tesserae and mortar with the bedding composition filling the interstices between tesserae.

As the placement of tesserae demanded a fresh lime bedding mortar, it was necessary to plan the dimension of the bedding mortar panels as a function of the capacity to fix the tesserae during the drying time. Thus, each panel represents a work planning unit, and in some mosaics, the boundary between these panels can be observed.

The tesserae were placed according to previously defined patterns, figurative or geometric, and were drawn through incisions in the bedding mortar; some of the imprints were captured under some mosaics (Figure 3a). These geometrically etched guides, called sinopia, were made with simple instruments that formed part of the common masonry tools: Rulers, calibers, compasses, nails, and ropes. Once the general pattern was established (Figure 3b), the specific drawings were painted with chalk or charcoal, whereas for the repetitive motifs that formed part of the decoration of borders and edges, templates could be used.

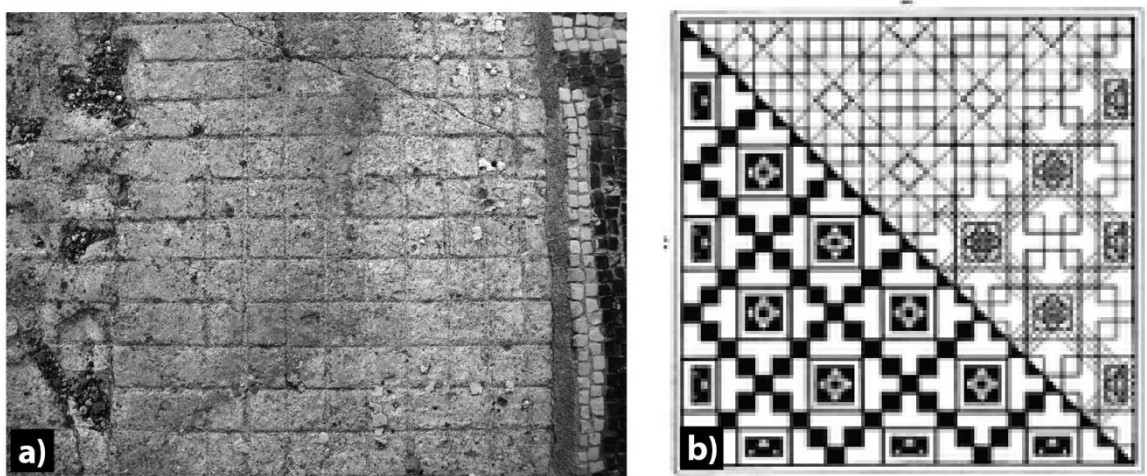

**Figure 3.** (**a**) View of the sinopia of a mosaic belonging to the domus of [21], from Pompeii. (**b**) Scheme of mosaic construction based on the sinopia, from [22].

According to [23], the pavements examined in past and modern archaeological excavations do not necessarily display the stratigraphy described in the ancient treatise of Vitruvius or others, as in [24]. Frequently, the thickness and the nature of their constituent materials vary among sites or even within the same building among pavements, which justifies investigations accounting for the whole structure of the pavement.

*2.2. Surface Analysis of the Studied Mosaic*

The studied mosaic is dated from the early 3rd century A.D. [3]. The mosaic covers the northern corridor of 15.5 m × 3.0 m, which provides access to the pars urbana (Figure 1). This floor was repaired during the last conservation intervention, which was carried out by the University of Évora [7], and this intervention consisted of vegetal species removal and surface cleaning with water and biocide application by means of nylon brooms.

The mosaic is formed by a decorative geometric composition of large octagons (Figure 4), outlined with a black filet over a white background. Three distinctive geometric compositions fill the inner octagons. At the ends of the corridor, there are defined panels (5 m in length each) formed by colored squares and diamonds on the same white background, and this pattern is repeated in all octagons; in the central zone (5.5 m), the octagons are filled with rosettes that are the same colors as the ends panels, creating a beautiful decorative effect. The composition is completely enveloped by a simple guilloche border produced with a double row of gray tesserae over the same white background [18]. The same constituent elements and forms are found in the two mosaic fragments of Merida [25], which leads to the assumption that the same model was used for the mosaics at both sites.

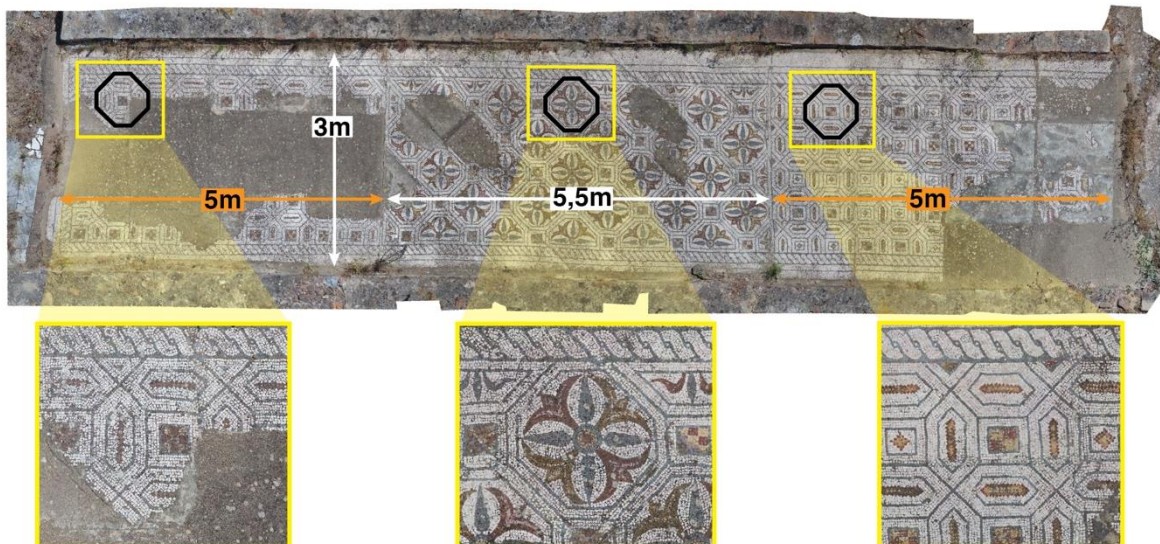

**Figure 4.** Photogrammetric orthomosaic of the entire studied roman mosaic (top), where the large octagons that define the geometry of the background pattern are shown. The bottom panels show a zoom in of the geometric compositions that fill the inner three sections of the mosaic.

### 2.3. Unmanned Aerial Vehicle (UAV) Imagery Acquisition and Processing

As a first product of the noninvasive analysis, a UAV was used to survey all of the Roman Villa of Pisões exposed area. The orthophoto and digital surface model (DSM) generated were used to map the area and provide a reference for further excavation, management, and monitoring works. A DJI Phantom 4 Pro UAV was used to acquire RGB imagery and Agisoft Photoscan was used for processing. Geodesic accuracy of the final products was possible by using five ground control points (GCP's). Four of these ground control points were temporary ones, placed in the corners of the surveyed area, and built on big screens with a pattern and a cross to be easily seen in the UAV imagery. Another GCP was collected approximately in the middle of the area, using a point in the mosaic pattern easily seen in the photos. The methodology used for this first task is the one detailed in [26]. The GCP's were collected using a Spectra Precision Epoch 50 GNSS receiver configured with base and rover antennas. Acquisition was made using post-processing kinematic (PKK), with further coordinate correction made with data from the fixed station of Beja, which belongs to the ReNEP network (GNSS National Network of Permanent Stations), which is the Portuguese service for geolocation based in fixed stations. High resolution images of these products, in jpeg format, can be downloaded from http://www.dct.uminho.pt/pisoes/ORTHO_HD_PISOES_GENERAL.zip (orthomosaic) and http://www.dct.uminho.pt/pisoes/DSM_PISOES_GENERAL.zip (DSM).

To detail the survey of the mosaic area that was studied, close-range photogrammetry was used, and a high-resolution orthophoto image and DSM were obtained for the mosaic surface.

The imagery data and final georeferenced products were also obtained by using a UAV and photogrammetric processing. The UAV used in this study (DJI Phantom 4 Pro) was equipped with a 20 megapixel camera, a 1 inch sensor, and a mechanical shutter. The camera was stabilized with a 3 axis gimbal. To capture the aerial data, two individual flights at low altitudes (~1.5 m) were conducted using manual control because there is no automatic flight control software that can be used at such low altitudes. The application DJI Go 4 was used to configure the UAV and to monitor and control the flight. This UAV had an obstacle avoidance system, which the manufacturer calls "Vision System". This system is based on three optical sensors (to frontward, backward, and downward obstacle avoidance), an ultrasonic sensor (as a redundancy to downward obstacle avoidance), and two infra-red sensors for leftward and rightward obstacle avoidance. The downward obstacle avoidance sensors were configured to 1.5 m distance to automatically avoid the UAV to get below that height.

Phantom 4 Pro can use the Vision System to accurately hover over a point with a vertical precision of ±0.1 m and a horizontal precision of ±0.3 m, according to the manufacturer. This position system was based on image analysis of the features below the UAV, obtained with the pair of small cameras that are also used for obstacle avoidance. This system was automatically used when both radio control sticks were put in the middle position. The UAV was then maneuvered very carefully with small shifts along the X axis, taking pictures when the UAV was completely still, until a complete stripe over the mosaic was completed. After a small shift along the Y axis was made and small shifts along X axis were again performed, pictures were taken. This time-consuming procedure ensured that all obtained pictures had enough overlap, both in X and Y axis, to be able to produce an orthomosaic and a DSM.

In total, 191 pictures were taken, covering 69.9 m$^2$ of area. A battery exchange was necessary to complete all the surveyed area using this technique. Figure 5 shows the position of each camera as well as the amount of overlap obtained.

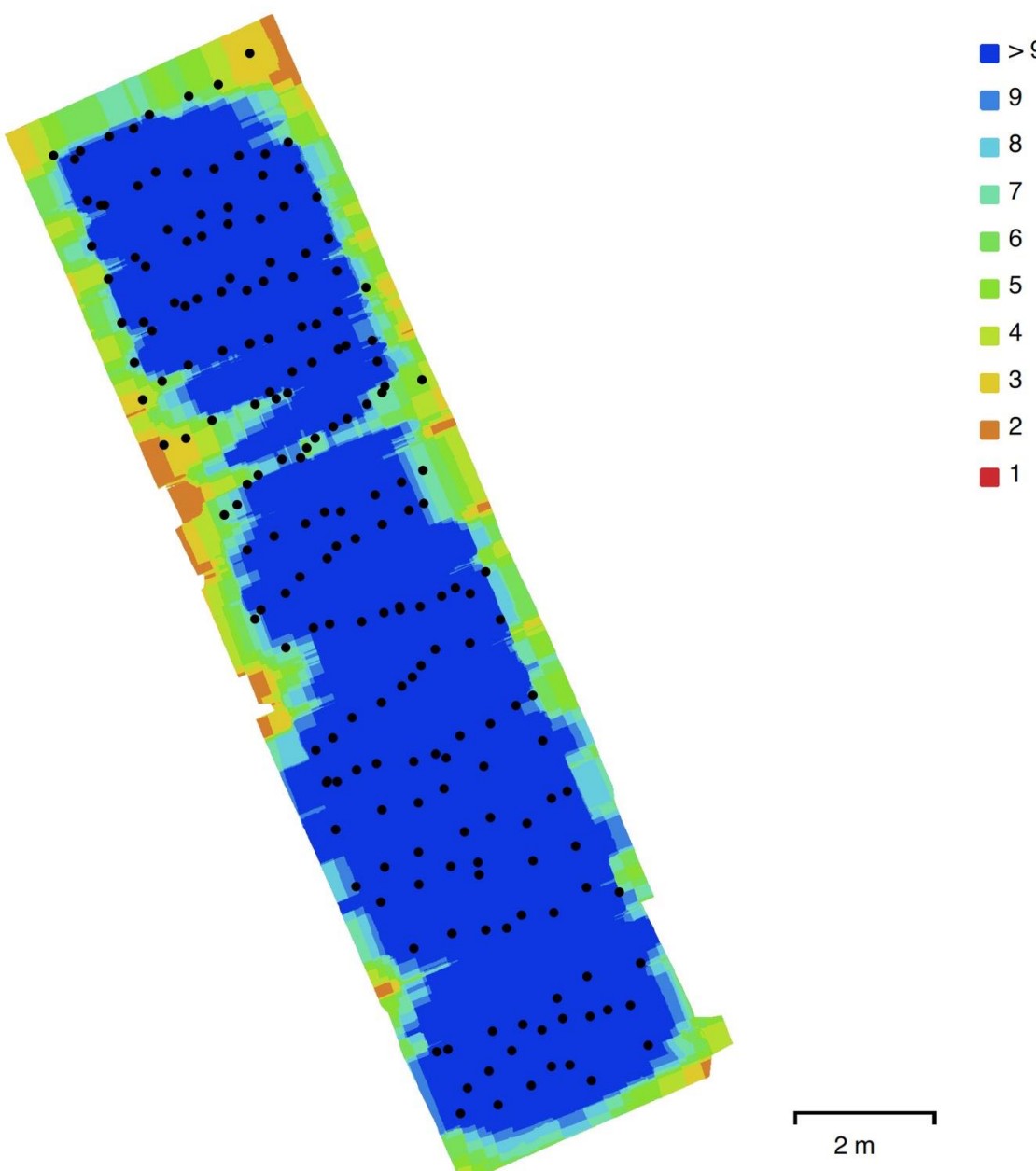

**Figure 5.** Camera locations (black dots) and image overlap. Colors represent the number of photos where each point appears.

The use of the UAV was chosen as opposed to a manual handheld camera because it allowed us to make a high-resolution orthomosaic of the whole mosaic extension without walking on it unnecessarily, and it was more practical to obtain perfectly vertical pictures from an elevated standpoint without too many shadows or obstructions, such as the photographer's own feet.

To perform the data processing, the first step was the photo alignment. The highest accuracy was used as well as the generic preselection and reference preselection parameters. In the advanced parameters, the key point limit was set to 60,000 and the tie point limit was set to 4000. Masks were used to exclude areas in the images where the UAV's shadow was present. Adaptative camera model fitting was also used. After this step, points with a reprojection error bigger than 0.50 were excluded from the sparse point cloud. In total 4305 points were excluded and 102,222 were maintained. After this step, a dense cloud with medium quality and a mesh was built, only to help in the placement of GCP's because they were, this way, placed automatically in each photo. To ensure the geometrical accuracy and georeferencing of the photogrammetry products, ten GCP's, which were mostly distinguishable features of the mosaic pattern, were used in the processing chain. The positions of these GCPs relative to the surveyed area are illustrated in Figure 6. The measurement of GCP coordinates was performed using a differential Spectra Precision Epoch 50 GNSS system. The average easting, northing, and altitude errors of the collected coordinates were 0.001 m, 0.002 m, and 0.010 m respectively. After the point cloud optimization, the resulting average errors were: 0.003686 m East; 0.002260 m North; 0.00988 m Altitude. The total RMS error was 0.004435 m and 0.415 pixels. The coordinate system used was PT-TM06/ETRS 89 (EPSG: 3763) which is the official coordinate reference system in Portugal mainland.

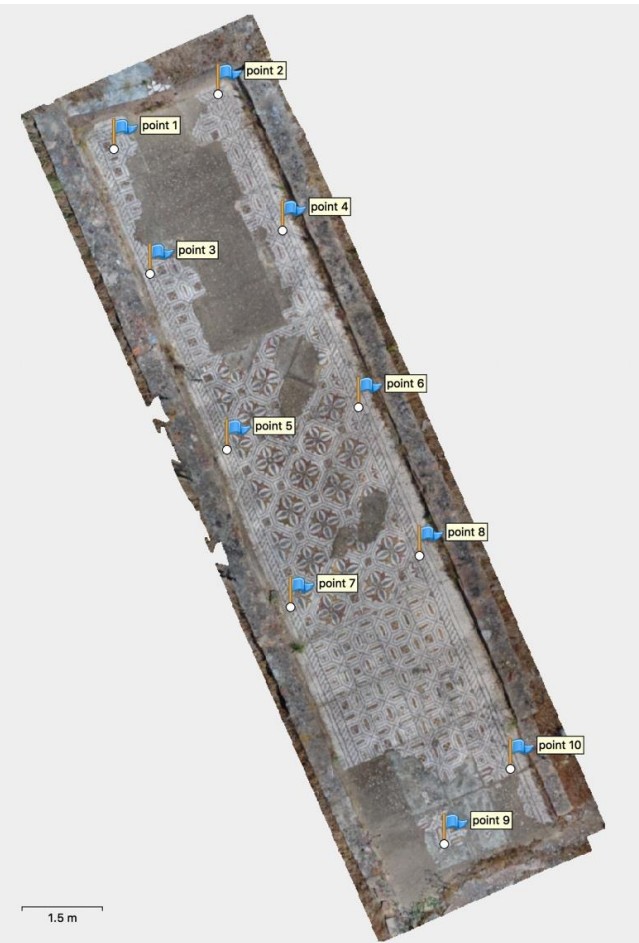

**Figure 6.** Position of the ten ground control points used to geometrically correct the unmanned aerial vehicle (UAV) mosaic survey.

After this step the dense point cloud was rebuilt using the ultra-high quality parameter and moderate depth filtering. A total of 442291309 points were generated. Although not necessary for this study, a mesh was also built using the dense cloud as source. The parameters used were height field (2.5 D) as surface type, face count set to high, and, in the advanced tab, interpolation was set to "enabled".

Finally, a very high-resolution digital surface model (DSM) and an orthomosaic of the mosaic area were generated. In both cases, all the parameters were the default ones. The source of data for the DSM was the dense cloud and for the orthomosaic was the DSM. The calculated ground sampling distance (GSD) of the products was 0.47722 mm. The results of the mosaic survey can be downloaded, in high resolution jpeg format from http://www.dct.uminho.pt/pisoes/DSM_PISOES_MOSAIC.zip (DSM) and http://www.dct.uminho.pt/pisoes/ORTHO_PISOES_MOSAIC.zip (orthomosaic). Some small portions of the DSM near the walls appear miscalculated due to the presence of moving vegetation, as can be easily seen in the orthomosaic. The photogrammetry processing using "structure from motion" is a very time consuming task that usually requires important hardware features to process data in the best quality possible. For all the photogrammetric processing in this work the computer used was an Apple MacPro with a Xeon E5 octacore 3 GHz processor, 128 GB of RAM, one SSD with 1 Terabyte disk, two internal AMD FirePro D500 graphical processing units and two external Nvidia GeForce GTX 1080 Ti graphical processing units (capable of 11.34 teraflops of processing power each).

*2.4. Inner Analysis of the Studied Mosaic*

2.4.1. Data Acquisition and Processing

GPR uses electromagnetic waves generated by an antenna moving on the surface. These waves penetrate in the subsurface and a part of their energy is reflected on the materials composing the structure, returning to the surface where they are detected on another antenna. The results are radar sections (radargrams) in which the *x*-axis represents the antenna survey line and the *y*-axis defines the propagation time of the radar wave in your path. In the case of floors and walls, a 1600 MHz antenna was used because it allows distinguishing reflectors of up to 50 cm inside. In the case of a 400 MHz antenna the investigation range is 3 m (approx.)

The selected mosaic portion has been recognized on two sensor resolution scales; first, a 2D-GPR survey was carried out using a 400 MHz antenna (yellow rectangle in Figure 7, lower image). These profiles were placed every 50 cm (approx.) along the width of the mosaic, and their lengths exceeded the dimensions of the study area in order to obtain a general vertical inner view. Second, we used a 1600 MHz antenna to conduct a 3D-GPR survey based on parallel profiles spaced 0.05 m apart (black rectangle in Figure 7, lower image) to obtain detailed volumetric images of the mosaic. The most significant acquisition parameters for both surveys are presented in Table 1.

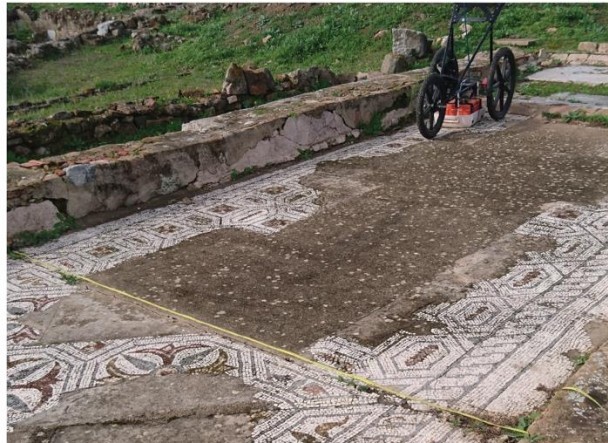
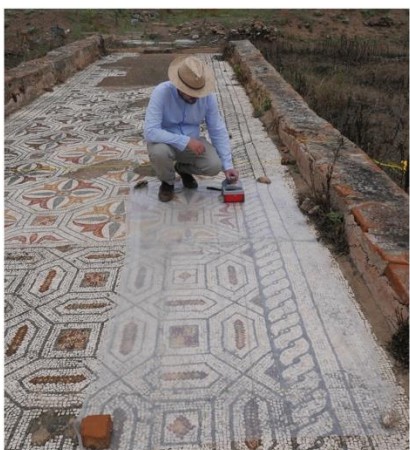
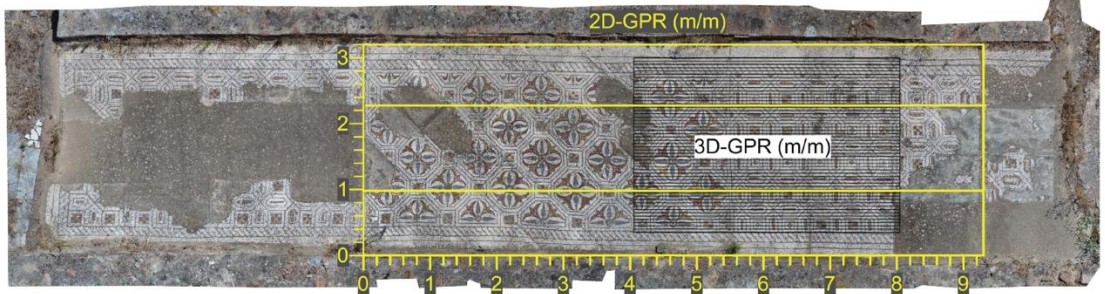

**Figure 7.** The upper left image shows the 400 MHz antenna, which was used in this study to generally inspect below the mosaic. In the upper right image shows the 1600 MHz antenna, which was used to conduct a detailed 3D inspection of the mosaic. The lower image corresponds to the orthomosaic of the corridor where the inspected rectangles were placed; in yellow, the area of the 2D ground penetrating radar (GPR) profiles; and in black, the 3D-GPR area with parallel profiles spaced 0.05 m apart.

**Table 1.** Most significant acquisition parameters.

| Antenna | 400 MHz | 1600 MHz |
|---|---|---|
| Time range | 60 ns | 10 ns |
| Scans/m | 50 | 800 |
| Samples/trace | 512 | 256 |
| Bit sample | 16 | 32 |
| Low vertical filter | 800 MHz | 32,000 MHz |
| High vertical filter | 100 MHz | 800 MHz |

The data processing was performed with RADAN-7 (GGSI, Inc.) commercial software, and for the 2D and 3D images, our own design codes were also used for presentation [27]. As the quality of the acquired signal in this study was good, conservative flow processing was applied (Table 2), which was mainly aimed at increasing the signal amplitude (gain corrections) and, most importantly, to increase the vertical and lateral resolutions (deconvolution operator). To determine the average dielectric permittivity, an exhaustive analysis of the velocities was performed by adjusting some diffractions in the 400 MHz antenna profiles.

**Table 2.** Most relevant flow processing steps.

| | |
|---|---|
| Zero correction | −0.23 ns |
| Gain correction | Constant in time window |
| Kirchhoff migration | V = 0.128 m/ns |
| Predictive convolution | N = 2.5, pred. Lag = 1.5 |
| Bandpass filter | 150–500 MHz |
| Gain correction | Linear in 11 time segments |
| Mean dielectric permittivity | 5.5 |

### 2.4.2. Methodological Aspects: Resolution and Feature Identification

The GPR device parametrization with the 1600 MHz antenna (Table 1) suggests a very high resolution, capable of inspecting objects with sizes compatible with the teasels (1–2 cm at 4 cm depth, Table 3). However, the GPR radargrams obtained from the Roman mosaic (Figure 8) did not allow for the high resolution suggested, although the radargrams clearly reveal the main stratigraphy of the mosaic.

**Table 3.** Absolute final accuracy for a 3D GPR grid.

| | |
|---|---|
| Lateral X resolution | 2–3 cm |
| Lateral Y resolution | 4–5 cm |
| Vertical Z resolution | "fuzzy" above 1.8 cm |
| | 4 cm at 4 cm depth |
| | 5.66 cm at 8 cm depth (top of *nucleus*) |

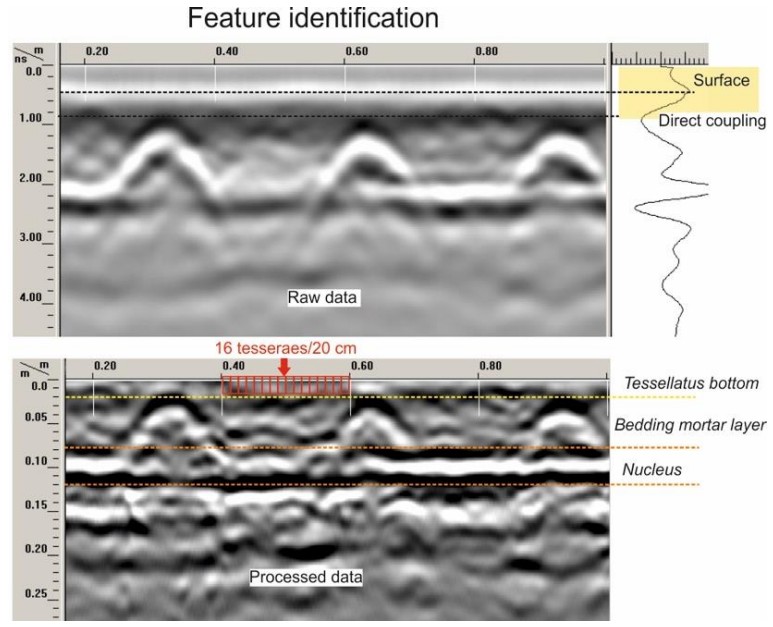

**Figure 8.** Fragments of acquired radargrams over the mosaic with the 1600 MHz antenna; the fragments contain the main identifiable features. The upper radargram is raw data, and the lower radargram is the same processed data, where we note that the 16 tesserae of the chain represented by the red grid, which fill 20 cm on average, are not detected. The three hyperbolas are caused by the sinopia main furrows.

The accuracy with which a specific body can be located depends on several factors. The GPR antenna begins sensing the target upon approach, continues to receive reflections as it passes over the target, and continues to receive reflections for some distance past the target. The distance between the antenna and the target changes as the GPR moves, which explains the hyperbolic shape of the reflection that determines the body detection.

In the first theoretical approximation, the GPR resolution is controlled by i) the bandwidth frequency of the electromagnetic pulses that radiate from the antenna in a cone of approximately 60 degrees amplitude and ii) the wave pulses velocity in the involved material. The two components that dictate the resolution [28] are the lateral resolution length, Δl, (sideways displacement) and the longitudinal resolution length, Δr, (depth), as depicted in Figure 9.

## Lateral and Longitudinal resolution

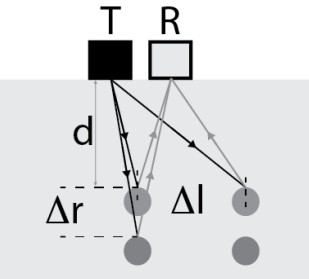

**Figure 9.** The GPR resolution can be divided into two parts: The lateral (Δl) and the longitudinal resolution (Δr).

The lateral resolution is described by the Fresnel zone equation:

$$\Delta l \geq \sqrt{\frac{d\lambda}{2}} = \sqrt{\frac{dv}{2f_c}} \tag{1}$$

where λ is the central wavelength, $v$ is the GPR wave velocity in the medium, $f_c$ is the central frequency and $d$ is the depth.

The longitudinal resolution, based on the Rayleigh principle, can be described by the following equation:

$$\Delta r \geq \frac{\lambda}{4} = \frac{v}{4f_c}. \tag{2}$$

If we assume an average velocity of 0.128 m/ns for the mosaic and 1600 MHz of central frequency, we have a limit of 2 cm for longitudinal resolution and 2–4 cm for lateral resolution at depths between 1 cm and 4 cm, respectively. This explains why the tesserae cannot be detected from data acquired with the 1600 MHz antenna, as shown in Figure 8.

There are other factors that also contribute to the final absolute accuracy:

(1) The positioning accuracy also depends on the sampling rates of the GPR signals, both spatially and temporally. In this study, we used a scan sampling of 800 scans/m and a time sampling of 0.04 ns between samples along each trace (Table 2); thus, we determined Nyquist values of 1.25 mm (1/2Δx) and 12.5 GHz (1/2Δt), respectively. Although these 800 scans/m were averaged in 4 traces (stack), the effective Nyquist number became 2.5 mm. These two sampling values were in accordance with the required resolution ranges, indicating that the acquisition parameters were appropriate in this study.

(2) The distance between the transmitter and receiver (T-R offset), which is often ignored in deep surveys, is comparable to the target depth in this study. For the 1600 MHz antenna, the T-R offset was 5.8 cm (GSSI, Inc.). This value defines a "fuzzy" top level in the raw data that can be made finer during the processing flow if we consider some signal features (Figure 8). The very first signal in a scan is called the "direct coupling" between the transmitter and receiver. This signal is used to identify the surface position in a scan. With the 1600 MHz antenna, the surface is located at the first positive (white) peak within the direct coupling. In our raw radargram, direct coupling looked like straight horizontal bands on top of the data window. The signal arrives at

the receiver before penetrating the material, which in this case, was after 0.4 ns or 4–5.25 cm if we consider the air-tesserae velocities. This signal was a combination of the transmit pulse in the air and surface reflection from the top of the material (top of mosaic), so the direct coupling carries little information about the top of the mosaic. However, the amplitude depends on the dielectric constant of the material, and variations in amplitude may indicate changes in properties (e.g., increased moisture). Direct coupling disguises the first part of the scan, and thus, making this signal as short as possible is a major configuration goal. For a 1600 MHz antenna, direct coupling allows for the detection of targets from 1.8 cm (5.8–4 cm) below the surface and the accurate measurement of their depths. Considering the negative peak (a straight horizontal black line in the radargram) immediately below the surface is part of the direct coupling. The first positive peak does not show any visible variations, although its amplitude may vary along the profile. Some variations may be observed within the negative peak, and they usually indicate changes in the layer properties within the top inch of the material, although their accurate interpretation is difficult.

(3)   Increasing the resolution with a spike deconvolution. When the target size is a fraction of the wavelength, the shape of the detection hyperbola does not change significantly. This means that any targets that are dimensionally less than 2 cm in length and 2–4 cm laterally will produce hyperbolas of the same size and shape. Relative sizing is possible for targets with a diameter greater than 2 cm, as long as these targets are located at the same depth, crossed at the same orientation, and surrounded by the same material.

These resolution limits can be reduced slightly in the data processing by applying a spike deconvolution [29]. In Figure 8, we compared the raw radargram with the processed radargram, observing a significant increase in resolution.

All of these previous considerations have been established for the 2D-GPR profiles (X, Z), but in our 3D survey, the (X, Z, Y) profile spacing was also considered. The orientation and number of profiles depends on the structure type and size, and other factors, such as distance control and wheel accuracy. The general rules are as follows: (a) The planned profiles could cross perpendicular to the features intended for detection. (b) A line spacing of 5–7.5 cm is required for complete coverage. This is the maximum practicable survey density that may be used for detailed 3D mapping because the 1600 MHz antenna width is 15 cm. (c) Linear targets that cross the survey lines at an angle of 45° to 90° will be resolved with good accuracy. (c) Complete exploration of the structure requires a survey in two directions, which are perpendicular to each other.

In this study, we planned a grid layout with the profiles spaced only 5 cm apart in the parallel direction to the corridor. By considering all the theoretical and experimental aspects that affect the GPR resolution, we could establish the absolute final accuracy, shown in Table 3. This means that the surface layer of the tesserae was unknown, and up to the nucleus level, only those elements that exceed the dimensions listed in Table 3 could be detected.

## 3. Results and Discussion

### 3.1. Surface Analysis Based on Photogrammetry

The orthomosaic (Figure 10a) consists of approximately 1 giga square pixel with a 0.5 mm ground sampling distance (GSD). It is a resolution that allowed each tesserae (of 1 cm$^2$) be defined of approximately 400 pixels. Figure 10b shows a grid of rectangular mesh of 1.2 m × 0.72 m oriented in two directions: (i) Along the corridor direction in the bottom sector, where the octagonal pattern is filled by squares and diamonds and (ii) oriented 45° with respect to the corridor direction in the upper sector, where the octagonal pattern is filled with rosettes.

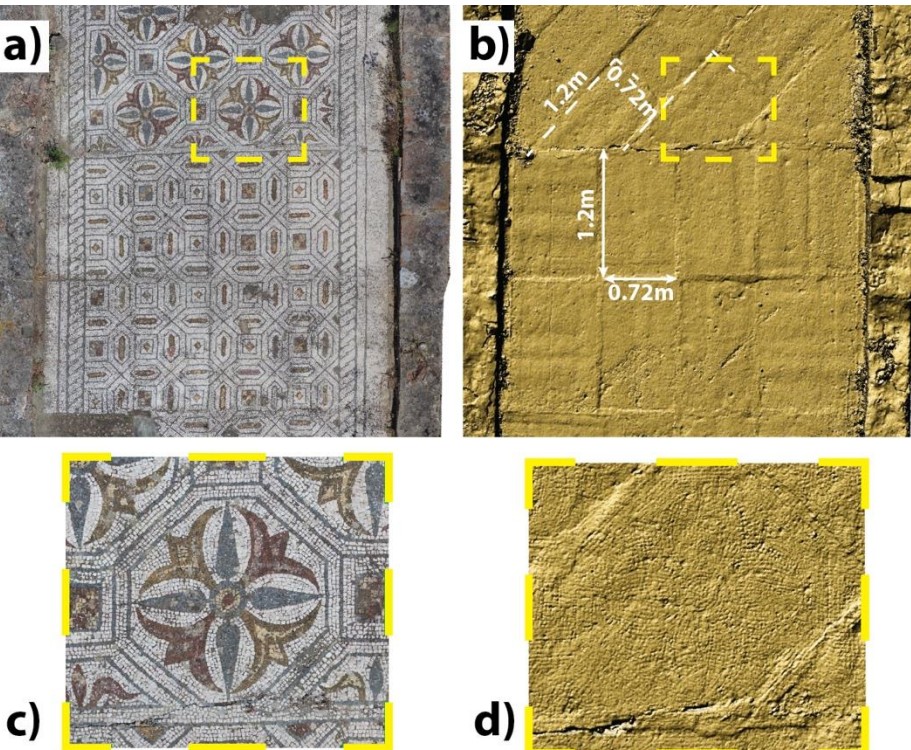

**Figure 10.** (**a**) Fragment of the obtained photogrammetric orthomosaic from the studied floor mosaic. (**b**) Fragment of the digital surface model (DSM) in the same area where the boundaries between the bedding mortar sections of different work planning units were perceived and measured. (**c**) Detail of the green detached square found in (a). (**d**) Detail of the green detached square found in (b).

Some in situ measurements of the various tesserae colors were made with a Vernier caliper and confirmed with a photogrammetric image. Table 4 represents these measures and the mean sizes, resulting in an average tesserae of 1 cm wide, 1 cm long. In addition, thickness measurements were made in 14 tesserae of the various colors, which were loose. These values ranged between 1.5–2.0 cm high. All of these measurements were considered when interpreting the results.

**Table 4.** Measurements of various tesserae to determine the average size.

| Tessera Types | | Measurements | | | | | | | | | Mean |
|---|---|---|---|---|---|---|---|---|---|---|---|
| White | Wide (mm) | 7 | 10 | 14 | 9 | 4 | 8 | 10 | 7 | 9 | 8.8 |
| | Long (mm) | 10 | 12 | 14 | 13 | 8 | 9 | 11 | 9 | 12 | 10.9 |
| Black | Wide (mm) | 7 | 9 | 11 | 10 | 7 | 11 | 12 | 12 | 7 | 9.6 |
| | Long (mm) | 12 | 9 | 14 | 12 | 9 | 13 | 14 | 13 | 9 | 11.7 |
| Dark red | Wide (mm) | 11 | 10 | 12 | 14 | 7 | 10 | 10 | 10 | 9 | 10.3 |
| | Long (mm) | 14 | 11 | 17 | 12 | 13 | 11 | 13 | 11 | 10 | 12.4 |
| Light red | Wide (mm) | 13 | 13 | 8 | 9 | 8 | 6 | 8 | 8 | 10 | 9.2 |
| | Long (mm) | 13 | 14 | 14 | 12 | 10 | 10 | 10 | 12 | 10 | 11.7 |
| Dark yellow | Wide (mm) | 10 | 10 | 9 | 12 | 12 | 8 | 10 | 9 | 8 | 9.8 |
| | Long (mm) | 11 | 12 | 9 | 12 | 12 | 10 | 12 | 10 | 11 | 11.0 |
| Light yellow | Wide (mm) | 10 | 6 | 12 | 12 | 10 | 7 | 9 | 9 | 11 | 9.6 |
| | Long (mm) | 11 | 8 | 14 | 12 | 11 | 7 | 13 | 10 | 11 | 10.8 |

In addition, the lineal and superficial densities of the tesserae have also been calculated by in situ measurements and over the orthomosaic. We can deduce that the lineal density of the tesserae for this mosaic is 15–16 tesserae/20 cm on average, and the superficial density is 112 tesserae/10 cm$^2$. Although the pieces are 1 cm long, the linear density decreases, given that the joint widths are unequal. In this context, the surface density is a very approximate average value because it varies according to the orientation of the tesserae that form the different drawings.

### 3.2. The Inner Mosaic Based on 3D-GPR

Figure 11 shows a portion of the raw 2D-GPR of the 400 MHz profile (lower left image) and two 2D-GPRs of 1600 MHz processed radargrams (upper images), which are coincident with the upper 2D-GPR profile (red rectangle). The right 1600 MHz radargram has been processed with the same flow, but a Kirchhoff migration has also been applied (marked step in Table 2).

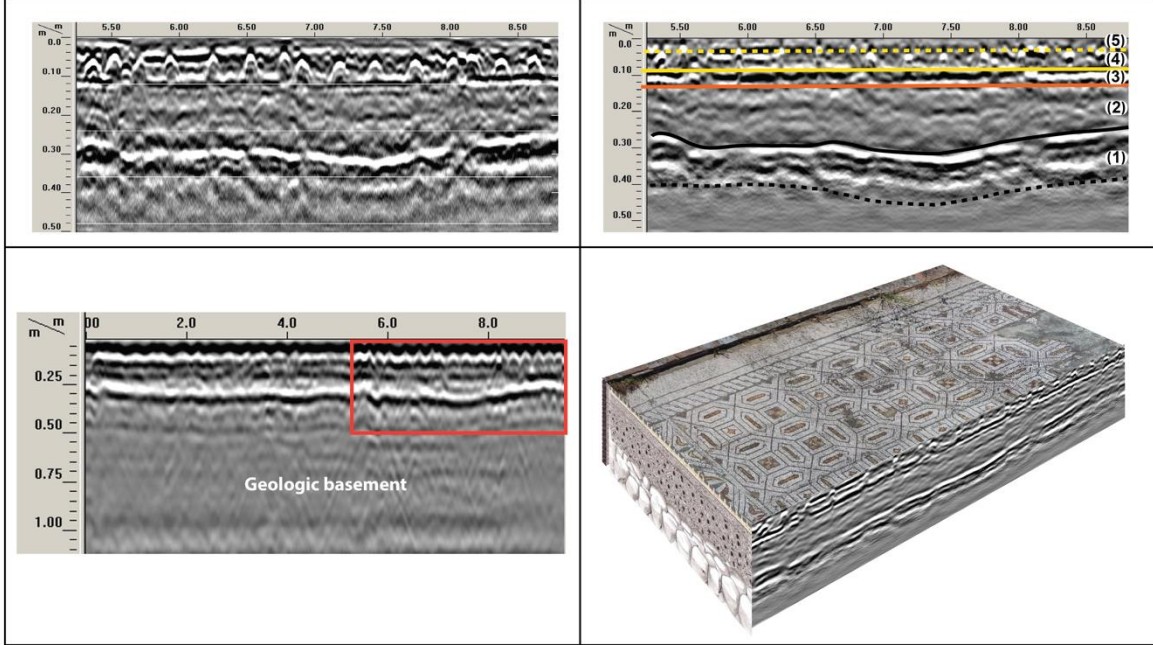

**Figure 11.** Top images show a 2D-GPR profile acquired with a 1600 MHz antenna, and the processing flow that was applied to this profile, as represented in Table 2. The right radargram differs from the left radargram because a Kirchhoff migration has also been applied. In both radargrams, we can distinguish all mosaic units (see the main text). The bottom left image is a raw 2D-GPR profile acquired with a 400 MHz antenna. The profile shows a general vertical stratigraphy view of the mosaic over the geologic bed. The red target indicates the location of the 1600 MHz 2D-GPR profile shown in the upper panel. The bottom right image is a general interpretation view of the main inner structure of the mosaic.

The first results of the 3D-GPR survey are shown in Figure 12, where the most significant detected levels up to the base of the nucleus are presented. From left to right, the two left images correspond to the photogrammetry: The one on the upper left is the orthomosaic and the bottom left is the DSM. The remaining images are horizontal cuts (depth slices) of the 3D-GPR model from the surface up to 15 cm depth, which contain the most reflective elements within this depth range [25]. The upper GPR slices in Figure 12 (0.8–1.8 cm and 2–4 cm) correspond to the distribution of the significant reflective elements in the bedding layer of the roman mosaic, which contains the tesserae. The black lines over the orthomosaic represent the interface between the blocks detected in DSM and the yellow lines are secondary guides corresponding to reflective elements detected on the GPR 3D slices up to 4 cm.

Inferred main synopias for the Roman floor mosaic

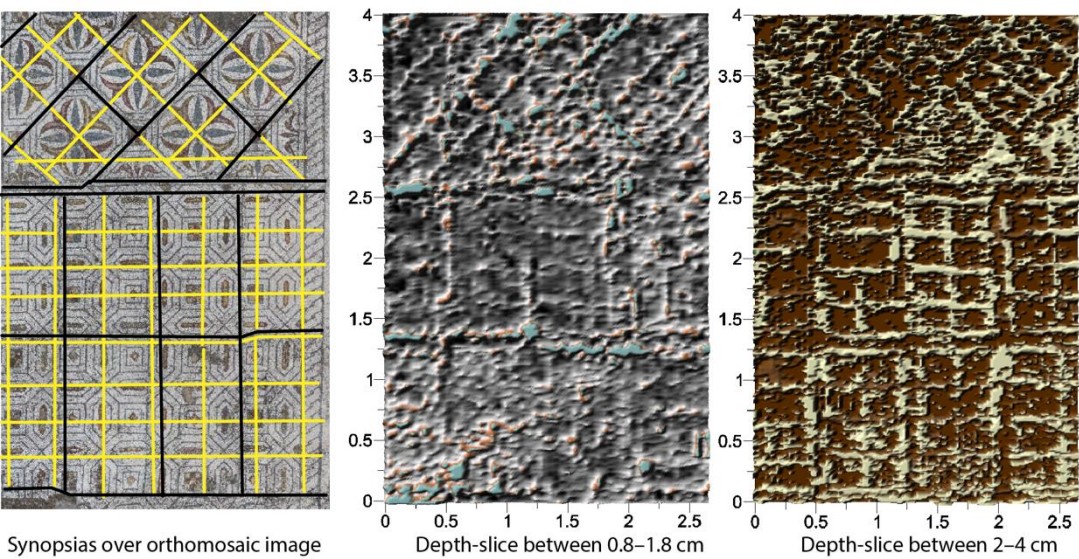

Nucleus layer for the Roman floor mosaic

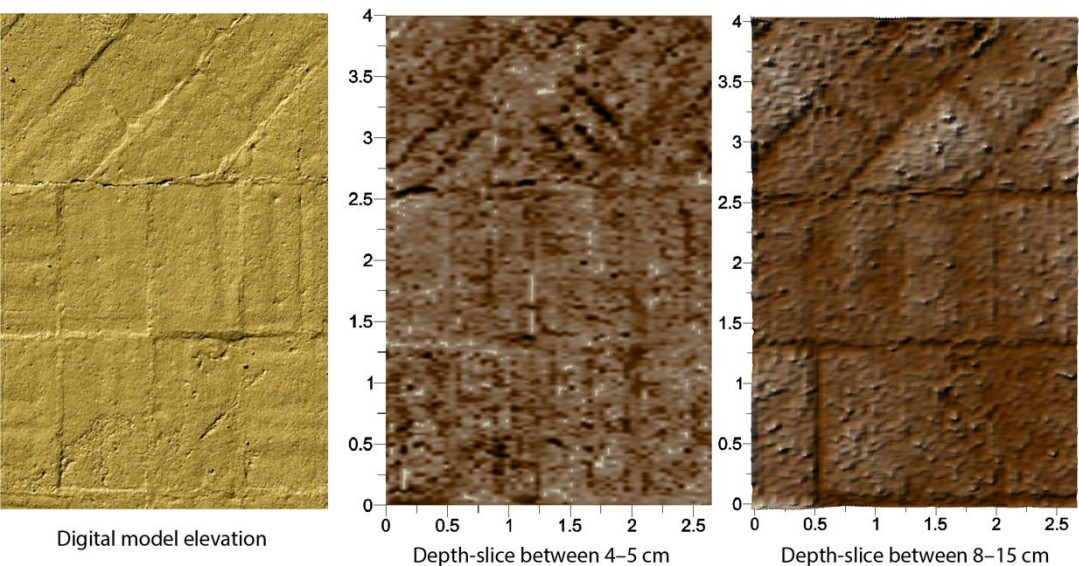

**Figure 12.** Upper images are the orthophoto (left) and two horizontal depth slices in the 3D-GPR model. Through the reflectors, we can infer the main sinopia of the Roman floor mosaic that is placed over the orthomosaic image (see discussion). Lower images are the digital surface model (DSM, left) and two horizontal depth slices cutting the nucleus: The first one is near the top of the nucleus, and the second integrates 7 cm of the nucleus body. Finally, the reflectivity correlates very well with the DSM model.

The lower GPR images in Figure 12 are the depth slice that correspond to the thin bedding layer of mortar (between 4–6 cm) just above the nucleus, where the sinopia are vanishing, and the slice that corresponds to the nucleus layer (between 8–15 cm).

### 3.3. Discussion

Figure 13 summarizes the detected levels up to the bottom of the rudus so that we have a complete study and documentation of the beautiful mosaic surface and its internal architecture.

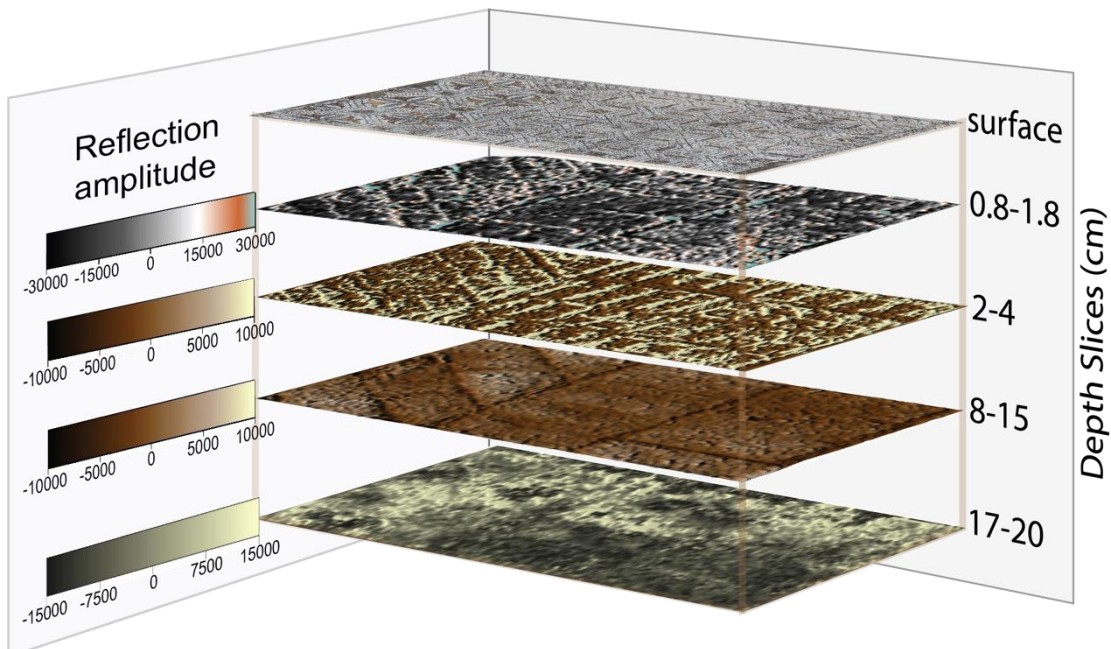

**Figure 13.** Summary of the most significant levels detected in the 3D-GPR survey, collected with a 1600 MHz antenna. The color scales correspond to the amplitude of the reflections.

### 3.3.1. Surface thorough Photogrammetry

The orthomosaic resolution allows a detailed inspection of the mosaic surface. The gigapixel digital surface model (Figure 10b) accurately captured the surface relief of the mosaic, where subtle changes, not perceptible with the naked eye in the field or in the orthomosaic, could be noticed on the DSM. For example, the relief that defines the grid in Figure 10b is in the order of 1.9 mm and was interpreted as the boundaries of the bedding mortar sections, corresponding to different work units, which dried up in different time windows.

Notably, we cannot take all the information from the orthomosaic regarding the marks of the drawing planning (orientation and placement of the tesserae), which would be inscribed on the soft mortar layer. This is because the photogrammetry data are optical and only describe the small irregularities of the external surface (top) of the tesserae. Therefore, to further investigate the mosaic construction, it was necessary to penetrate the mosaic, i.e., to look at the surfaces beneath (bottom) the tesserae. For this purpose, we planned to use a high-resolution GPR (1600 MHz) that interferes with these design marks and penetrates into the interior of the tesserae.

### 3.3.2. Inner thorough GPR

The GPR images in Figure 11 show the main stratigraphic parts of the mosaic, which suggest that one could attempt a 3D inspection to analyze the conservation aspects (humidity, internal fractures, etc.) to make some preservation determinations. The top two layers of Figure 11 (up to 4 cm depth) are more special because they contain information about the artistic construction of the mosaic, and these layers can reveal interesting aspects from the patterns that help to further the knowledge of the mosaic. Thus, the radargrams and lower image in Figure 11 allowed us to establish the first interpretations of the mosaic stratigraphy: (1) the statumen basement layer is between 32 cm and 42 cm depths (on average), and this layer is characterized by a strong reflector at the top followed by a bed of coarse gravels that were well identified on the radargrams; (2) the rudus is a homogeneous layer with hardly any reflections, placed between 12 cm and 32 cm depths; (3) the nucleus, which is characterized by a double strong reflector, is located between 8–12 cm; and for (4) and (5), we could distinguish the

tesserae level, composed of a bedding mortar layer up to 2–4 cm, where designs marks were made and the tesserae were inserted (see also Table 3).

As expected from the previous considerations about the resolution and features identification from GPR data, the first considered layer in the 3D GPR model (between 0.8–1.8 cm, Figure 12) was a "diffuse" image that contained the base of the tesserae (bottom of layer 5 of Vitruvius description) and a part of the mortar that filled the interstices between the tesserae and fixed them. At the following level, represented in Figure 12 (between 2–4 cm depths), a set of geometric reflectors were detected, from whose arrangement the basic sinopia marked in the mosaic construction was inferred. The black lines drawn on the top left image of Figure 10 mark the main blocks detected in DSM and the attenuate segments in 2–4 cm depth-slice; while the yellow lines are the secondary guides that only detect a reflector segments in 2–4 cm depth slice. In the section of the floral motifs, the element that articulated the drawing was a mesh formed by squares with 36 cm sides that were arranged at 45° with respect to the direction of the corridor. For the section of geometric motifs, the basic element was a rectangular mesh with cells of 37 cm × 29 cm parallel to the corridor. The reflections that correspond to the nucleus layer (8–15 cm in Figure 12) correlate perfectly with the DSM, suggesting that in this mosaic, the nucleus was built with rectangular pieces of 1.2 m × 0.72 m distributed parallel, and at 45°, with respect to the corridor direction. Below the nucleus layer, the reflections in the rudus did not present significant distributions, except changes in signal amplitude that could be associated with areas of higher humidity, as mentioned in the previous section. Figure 11 is a summary of the detected levels up to the bottom of the rudus.

## 4. Conclusions

Through this work we have presented a methodology to approach the study of mosaics from the point of view of their construction. Our objective was to demonstrate how the photogrammetry (UAV) allows us to document, with a submillimeter pixel resolution, the decorative image of a Roman mosaic; which is far below the tesserae dimensions (1 × 1 × 2 cm). In addition, the photogrammetry also allowed for obtaining a digital surface model of the Roman mosaic, also with a submillimeter sensibility in height, sufficient to detect the main units of the sinopia (base blocks), the superficial cracks, and the differential settlements of the subsoil.

On the other hand, the 3D-GPR survey with a 1.6 GHz antenna provided a tridimensional model of the mosaic interior with a resolution that made it possible to establish the main guidelines of the sinopia, the base blocks, and distinguish all constructive layers of the Roman mosaic (statumen, rudus, nucleus, and bedding); except the first undetectable tesserae surface layer.

For this reason, we propose that the best way to know a Roman mosaic is to cross both nondestructive techniques: The UAV photogrammetry to cover the most superficial information of the mosaic, and the 3D-GPR to access its internal parts, between the surface and approximately 30–40 cm depth. In addition to the structural aspects, it was also possible uncover the main grooves that define the sinopia design. Curiously, the study revealed that even in areas where the tessellatum failure had been changed to a regularizing mortar, the GPR pattern of the lower layers remains. The grooves in the sinopia continued to be visible on the radargrams, which proves that the mortar coating covers only the surface of the tesserae and did not interfere with the remainder of the mosaic structure or its preservation state.

The results of this study, which involved obtaining the internal structure of the mosaic, suggest that the model quality was achieved, particularly for the pathology diagnostics such as deformations, fracture zones, and humidity. The results of the present work did not clarify the humidity content because the studied mosaic is very stable and not affected by humidity, since the data acquisition was completed in July, which is an extremely dry season in Portugal. In any case, the methodology followed in this study allows for extensive diagnosis prior to preservation and restoration works, even in the lower layers of the mosaic structure, to prevent the increase of the pathological agents that promote its degradation.

In the future, we intend to carry out another study through multi-array GPR frequencies and recollect the data in two perpendicular directions to better resolve the accuracy reach ratios.

Finally, through this work, we showed that a Roman mosaic, in addition to the beautiful superficial drawings defined by the tesserae, is a complex archaeological structure. It is precisely the quality of this structure that allowed the mosaic to be preserved in all of its splendor to the present, nearly two thousand years after construction.

**Author Contributions:** Conceptualization T.T. and B.C.; methodology T.T., B.C., and R.H.; data processing T.T., R.H., and R.J.O.; data acquisition, R.J.O., J.A.P., R.H., T.T., J.F.B., and B.C.; original codes J.A.P., data analysis B.C., R.H. and T.T., writing—original draft preparation and edition, B.C. and T.T.; figures edition B.C., T.T. and R.H.; writing—review R.J.O., R.H., J.F.B., A.C., and J.A.P.; historical and archaeological specialist A.C.; supervision, B.C.

**Funding:** This work is co-financed by the INTERREG 2014-2020 Program, through the project, "Innovación abierta e inteligente en la EUROACE" with the reference 0049_INNOACE_4_E; by the European Union through the European Regional Development Fund, included in the COMPETE 2020 through the ICT project (reference UID/GEO/04683/2019), and FCT project SFRH/BSAB/143063/2018. This research study was carried out using instruments acquired thanks to European Regional Development Fund (ERDF).

**Acknowledgments:** To the Action Plan to the Roman Villa of Pisões of the Évora University that manages the villa, by the delivery of the infrastructure; to the Institute of Earth Sciences, for the logistic support in the management of the resources affected to the development of this work and to D. Conceição, guard of the villa, for the warm reception of the team.

**Conflicts of Interest:** The authors declare no conflict of interest.

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
