# Peer review of "Studying the Construction of Floor Mosaics in the Roman Villa of Pisões (Portugal) Using Noninvasive Methods: High-Resolution 3D GPR and Photogrammetry"

_remotesensing, doi:10.3390/rs11161882_

Round 1

Reviewer 1 Report

The paper describes an interesting interdisciplinary approach employing photogrammetry and 3D GPR to map and study the external and internal structure of a mosaic.

The paper is well written, stating the objectives  clearly and presentation the methodological aspects in compact and solid way. The results are convincing the the interpretation can be justified by the collected data and subsequent analysis.

The overall impact of the work in the archaeological community is considered important especially in the cases that specific restoration actions have to be planed.

The paper is recommenced for publication with subject to a small comment that lies on the need from the authors to present a literature review regarding previous similar works involving either the application of geophysics or the employment of photogrammetry in the specific subject.

Author Response

The paper describes an interesting interdisciplinary approach employing photogrammetry and 3D GPR to map and study the external and internal structure of a mosaic.

The paper is well written, stating the objectives clearly and presentation the methodological aspects in compact and solid way. The results are convincing the interpretation can be justified by the collected data and subsequent analysis.

The overall impact of the work in the archaeological community is considered important especially in the cases that specific restoration actions have to be planed.

The paper is recommenced for publication with subject to a small comment that lies on the need from the authors to present a literature review regarding previous similar works involving either the application of geophysics or the employment of photogrammetry in the specific subject.

Thanks for your suggestions. In accordance with you we have incorporated some literature review references for the GPR and Photogrammetry in the Roman mosaics (lines 90-98 of the reviewed version).  

Reviewer 2 Report

Before acceptance, the paper must be improved especially when presenting the methods and partly the results.

In particular the photogrammetric survey is not well described, so it is not clear the accuracy (wanted/achieved), the overlaps, (why 32 flights for such a small area?), where are the GCP located, how many, the accuracy of their position, how was the RTK and PPK performed, which solution has been used, the processing parameters of Photoscan etc. You need to integrate a lot this part.

GPR survey: looking at fig. 5 it seems that only 3 profiles were measured, can you say the width of the GPR device (to compare with the overlap of 50 cm...)?

page 11: how where the tesserae measured in situ?

fig. 8 b and 8 d and 10 d(?) are almost unreadable because of the small contrast among the colors.

Fig 10a (?) what's the meaning of the green and purple lines?

You need to summarize the results obtained fro the two approaches and how they are integrated.

Author Response

Before acceptance, the paper must be improved especially when presenting the methods and partly the results.

In particular the photogrammetric survey is not well described, so it is not clear the accuracy (wanted/achieved), the overlaps, (why 32 flights for such a small area?), where are the GCP located, how many, the accuracy of their position, how was the RTK and PPK performed, which solution has been used, the processing parameters of Photoscan etc. You need to integrate a lot this part.

Thank you for the appropriate questions. To answer to all the issues mentioned, the text was extensively edited to have a better description of the UAV survey and data processing (Subsection 2.3; lines 208-318 of the reviewed version). The “32 flights” was a typo error. Only two surveys were made resulting in 191 photos. Two surveys were necessary because of battery exchange. Links to download high resolution survey results (in jpeg format) were added for to the article’s text for readers access and quality checking.

GPR survey: looking at fig. 5 it seems that only 3 profiles were measured, can you say the width of the GPR device (to compare with the overlap of 50 cm...)?

Yes, in fig. 5  (Figure 7 in the reviewed version) only 3 2D profiles with a 400 MHz antenna are made. We make only these 3 profiles in order to obtain general views of the mosaic (stratigraphy) in order to compare “vertically” with the 1600 MHz profiles; as it is show in Fig. 11 (of the reviewed version). 

Sorry but we do not understand the overlap of 50 cm. We only make 3 GPR profiles (2D) taking in to account the width of the mosaic and space them 50 cm aprox. In order to obtain a regular vertical views, but with a 400 MHz antenna profiles we don’t make a 3D GPR set.

Your comment indicates that the text is not understood and we have modified it.

Page 11: how where the tesserae measured in situ?

The measures of tesserae have done in two ways:

1)    First we measure them in situ with a Vernier Caliper (see photo)

2)    Second, we verify these dimensions on photogrammetric images. 

We have specified this in the text

Fig. 8 b and 8 d and 10 d(?) are almost unreadable because of the small contrast among the colors.

We've changed the colors to increase the contrast, thank you. 

Note: in the in the reviewed version fig 8 and fig 10 wave changed to Fig 10 and 12 respectively

Fig 10a (?) what's the meaning of the green and purple lines?

The green and purple lines are the sinopia divisions at second order, and the black lines coincide with those marked on the digital elevation model of Fig. 10 (version reviewed).

As it is not clear, we have modified the text, the Figure 12 and the figure caption of Figure 12.

You need to summarize the results obtained for the two approaches and how they are integrated.

Thanks for the suggestion, we agree. Was incorporated a new paragraph in conclusions concerning this issue (lines 740-753 of the reviewed version)

Reviewer 3 Report

Taking the Roman Villa of Pisoes as the study site, in this study, the advantage of noninvasive remote sensing approaches (the combination usage of GPR and UAV photogrammetry) was vividly demonstrated. The mm-level DSM as well as the subsurface tomography of mosaic structure dating back to Roman times were derived, which can be significant for the pathology diagnostics and sustainability conservation of those relics. The manuscript can be publishable in particular after considering the following comments:

1) Abstract, typo of 1.6 MHZ, it should be 1.6 GHZ or 1,600 MHZ.

2) Line 190-191, more technical issues linked to the UAV data acquisition and data processing is need. Firstly, manual contril for ~1.5 m of the UAV is challenging, how the variation of flight height, such as with an error of 0.5 m, will impact the precision of orthoimage and DSM product?

3) Line 250-280, authors interpreted that the tesserae (with a length of 20 cm, as illustrated in Figure 6) is caused by the 2-4 cm of lateral resolution. Theoretically, it is not true because the mentioned resolution is clearly better than the length of the unit observed. I think, the fuzzy top level of radargram (T-R offset) can be the main cause. Please add more clarification on this issue.

4) Line 315, typo error of Discursion

5) Figure 9. The vertical axe of GPR profile obtained in 400 MHz should be transformed into meter in order to keep consistent with 1.6 GHZ profiles.  

Author Response

Moderate English changes required 

The paper was reviewed by the Nature Research Editing Service

1)    Abstract, typo of 1.6 MHZ, it should be 1.6 GHZ or 1,600 MHZ.

Of course! Thank you

2)    Line 190-191, more technical issues linked to the UAV data acquisition and data processing is need. Firstly, manual contril for ~1.5 m of the UAV is challenging, how the variation of flight height, such as with an error of 0.5 m, will impact the precision of orthoimage and DSM product?

Thank you for the appropriate questions. A better description of the UAV survey and processing data was added to the text to answer to all the issues mentioned (lines 207-317 of the reviewed version). The 1.5 flight is challenging and must only be done by a well-trained pilot. Otherwise there are risks of crashing the UAV. At this close range, even if there were vertical variations of 0.5m (or one third of the average flight height) there would not be any major impact in the final precision of the orthoimage and DSM products. At this close distance, possible variations induced even by 0.5 m height differences are sub-centimetre ones (the difference between the highest and lowest altitude photos in the survey was about 33 cm). Software packages that are able to process close-range photogrammetry, such as Agisoft Photoscan, can easily solve camera positions and make a very precise 3D reconstruction using bundle adjustment algorithms or “Structure from motion” if photos are taken with a good overlap (between 70 and 90%) which was the case in this work. Moderate height differences are perfectly tolerable and do not have any major impact in the desired Ground Sampling Distance (in this case we achieved a GSD of 0.477 mm/pixel). Links to download high resolution survey results (in jpeg format) were added to the article’s text for readers access and quality checking.

3)    Line 250-280, authors interpreted that the tesserae (with a length of 20 cm, as illustrated in Figure 6) is caused by the 2-4 cm of lateral resolution. Theoretically, it is not true because the mentioned resolution is clearly better than the length of the unit observed. I think, the fuzzy top level of radargram (T-R offset) can be the main cause. Please add more clarification on this issue. 

In Fig.8 (of the reviewed version) we illustrated a set of 16 tesserae with a 20 cm length according with the generic dimensions of tesserae in Table 4. This is a direct measure. 

In other hand, in both radargrams of Figure 8 they are 3 visible hyperbolas that corresponding to the sinopia furrows. We make this in the figure caption of Fig.12

After the weighing of the two radargrams we have proceeded to make a calculation of the resolution provided by a 1.6 GHz antenna with the aim to demonstrate that the tesserae cannot be detected one by one using a 1.6 GHz antenna. 

First we have calculate the theoretical resolution (eq. 1 and 2) and on this resolution we have considered other instrumental aspects that add up to obtaining the absolute final accuracy of GPR survey. That is presented in Table 3.

4)    Line 315, typo error of Discursion

Thank you

5) Figure 9. The vertical axe of GPR profile obtained in 400 MHz should be transformed into meter in order to keep consistent with 1.6 GHZ profiles.  

It has been changed

Note: in the in the reviewed version fig 9 was changed to Fig 11